# Influence of Curvature Feature on Laser Heating during Tape Placement Process for Carbon Fiber Reinforced Polyether Ether Ketone Composite

**DOI:** 10.3390/polym15020289

**Published:** 2023-01-06

**Authors:** Hongquan Liu, Yong Li, Dajun Huan, Wuqiang Wang, Yanrui Li, Lisha Li

**Affiliations:** College of Materials Science and Technology, Nanjing University of Aeronautics and Astronautics, Nanjing 210016, China

**Keywords:** thermoplastic composites, laser material processing, power density distribution, Super-Gaussian profile beam, power density distribution, scattering behavior

## Abstract

The curvature feature makes the irradiance and absorptivity change, resulting in an uneven power density distribution, which affects the quality of composite parts. In this study, a theoretical model-based Super-Gaussian profile beam in the laser irradiation area was established to obtain the heat flux distribution on the curved surface. The effect of curvature on the surface scattering reflection, temperature distribution, and surface morphology were investigated and verified the validity of the theoretical model. Furthermore, the influence of the laser intensity distribution, laser inclination and curvature radius on the power density distribution and distribution uniformity were studied. Research indicated that the power density increases as the distance from the origin increase resulting from the variation of the irradiance and absorptance along the circumference. The flatter the intensity distribution of the laser beam in the height direction, the less uniform the power density distribution. Accordingly, the typical Gaussian profile beam significantly ameliorates the power density distribution. This research provides a novel understanding of using heat sources during laser heating thermoplastic tape placement.

## 1. Introduction

Laser-assisted automated tape placement (LATP) is a rapid additive manufacturing method for high-performance thermoplastic composites that offers new possibilities for the efficient and high-quality production of composite parts. As an important high-performance thermoplastic composite material, carbon fiber reinforced polyether ether ketone composites (CF/PEEK) have been widely used in tape placement. This method is appealing in aerospace applications because it takes full advantage of the benefits of thermoplastic composites and additive manufacturing. The in situ consolidation process eliminates the need for lengthy and costly autoclave cycles [1], dramatically reducing manufacturing time and costs. Because it is challenging to meet quality and production criteria, the in situ consolidation process has not taken advantage of this potential.

The deposition of layers of thermoplastic composite prepreg tape onto a mould or tool by a moving placement head is the basic principle of LATP processing. An energy source heats the new layer and the substrate before they contact the nip point under a compaction roller during the process. Maintaining the temperature within the processing window is critical for the final quality of the CF/PEEK composites component. LATP is a complex process involving various interactions, such as optics, heat transfer, phase change, polymer molecular chain movement, and molecular chain healing. Bond quality [2,3,4,5,6,7], crystallinity levels [8,9], void content [10,11,12], and the possibility of thermal degradation are all strongly influenced by the temperature history throughout the process. Consequently, the temperature in the beam irradiation zone is crucial. Nevertheless, the compaction roller during the lay-up process causes the income tape to have curvature characteristics. The tilted beam leads to a laser beam spot elongation [13,14] and incident angle variation, resulting in non-uniform laser power density distribution on the surface of the income tape. A shadow zone is formed before the nip point [15,16,17]. Specifically, the curvature of the tape and flat substrate is considerably different, impacting power density uniformity in the laser beam irradiation region.

Currently, most studies assume a uniform power density distribution and consider the CF/PEEK tape surface as an ideal surface with complete or partial absorption [18,19,20,21], failing to mention the anisotropic reflection behavior of composites. Some studies have been conducted to describe the reflection and absorption of the laser beam by developing an optical model during the lay-up process to obtain the heat flux distribution. The authors of [22] established an optical model based on a two-dimensional ray-tracing method of specular reflection, considered the effect of beam reflection on the heat flux distribution, and verified the validity of the model by utilizing the surface temperature distribution obtained by laser assisted placement experiment, which inadvertently demonstrated the validity of the optical model. In [16,23], a micro-semi-cylindrical model was created to describe the non-specular reflection behavior of the CF/PEEK composite. This model was then used to simulate the anisotropic scattering behavior of the CF/PEEK composite. These optical models primarily apply to substrates with a flat surface. BRDF is used to describe the relationship between the incident and reflected light. [24,25] established a 3D non-specular reflection model based on the bidirectional reflection distribution function (BRDF) [26,27] and calculated the heat flux distribution on the surface of the curvature mould and income tape based on the theoretical model during the laser-assisted winding process. Furthermore, [28,29,30] investigated the heat flux distribution on the surface of curvature and flat specimens during the laser-assisted filament winding process. Remarkably, these studies are based on the optical model during the winding process and do not account for the irradiation differences between the income tape and the flat substrate. Additionally, the power density distribution of the curved income tape remains to be investigated. Consequently, it is essential to investigate the influence of curvature on the interaction between the laser beam and the CF/PEEK composite during laser-assisted tape placement and to optimize the heat flow distribution to minimize the difference between the curved income tape and flat substrate for laser heating lay-up.

In this paper, the effect of curvature features on the reflection behaviour, heat flow distribution, and microstructure of the surface of the laser beam irradiation area is investigated during the laser heating lay-up process, and the interaction cross-section of the laser beam on the income tape is obtained to contribute to offering a fundamental basis for the subsequent improvement of the temperature distribution. Initially, based on the bidirectional reflectance distribution function scattering model, the influence of the laser beam inclination angle on the reflection behaviour of the CF/PEEK composite was investigated. Subsequently, based on energy conservation and Schlick equations, a theoretical model for the laser beam distribution on the surface of curvature tape is developed. Eventually, the optical-thermal coupling model and laser heating curvature tape experiment were conducted to verify the validity of the theoretical model. The effects of beam cross-section intensity distribution, laser beam inclination angle, and curvature radius on the heat flux distribution are investigated.

## 2. Materials and Methods

### 2.1. Material

The thermoplastic composite tapes used for this study unidirectional carbon fiber reinforced polyether ether ketone towpreg VESTAPE^®^, supplied by Evonik Industries AG. The UD tape utilizes a standard modulus carbon fiber (HTS 45) with a 60% fiber volume content and Evonik VESTAKEEP^®^ PEEK matrix. The thickness and width of the tape are 0.16 ± 0.1 mm and 12.7 mm, respectively. The material constant of CF/PEEK is listed in Table 1.

### 2.2. Scattering Reflection Measurement

Typically, the optical scattering behavior of the material is described using a bidirectional scattering distribution function (BSDF), which reflects the intensity correspondence between the incident and emergent radiation, such as the bidirectional reflection distribution function (BRDF) [27], the bidirectional transmission distribution function (BTDF), and the bidirectional diffraction distribution function (BDDF). The reflective optical behavior of the composite could be described by employing the optical micro-model in [31], and it was found that most of the energy was absorbed and reflected by the first row of fibers on the surface of the carbon fiber reinforced composite, and the fibers below the surface layer (at a depth of about 10 μm). There was a small effective incident depth of the laser beam compared to the carbon fiber reinforced composite prepreg tape [23]. Consequently, the bidirectional reflection distribution function (BRDF) is applied to characterize the optical scattering behavior of the CF/PEEK composite. BRDF is the ratio of irradiance scattered by a surface in a given direction to the irradiance incident on that surface alone [27], as shown in Figure 1, and is expressed as follows:(1)$$f_{r}\left(\theta_{i},\phi_{i},\theta_{r},\phi_{r}\right)=\frac{dL_{r}\left(\theta_{i},\phi_{i},\theta_{r},\phi_{r}\right)}{dE_{i}\left(\theta_{i},\phi_{i}\right)}$$
where *dL_r_*(*θ_i_,ϕ_i_,θ_r_,ϕ_r_*) is the reflected spectral intensity in direction (*θ_r_*,*ϕ_r_*), *dE_i_*(*θ_i_*,*ϕ_i_*) is the irradiance in the incident direction (*θ_i_*,*ϕ_i_*).

It is demonstrated that the curved tapes influence the incident angle, which ranges from 37° to 75°. Therefore, it is essential to investigate the effect of incident angle on the interaction between the laser beam and the CF/PEEK composite. Gonioreflectometer measurements could be carried out to investigate the reflection intensity distribution from non-homogeneous surfaces [25,32,33], such as the tapes considered in this study. Understanding the complex relationship between fiber orientation, incident angle, and non-specular reflection patterns is crucial. The anisotropic reflection behavior of the CF/PEEK composite prepreg tape was measured using the BRDF absolute measurement system, as shown in Figure 2. Depending on the actual incident angle variation for income tape, the following incident angle variations were considered: 15°, 30°, 45°, 60°, 75°, and 85°. The complicated intermediate computational procedures, such as scattering, absorption, and reflection of light in the prepreg tape, can be omitted by importing the measured BRDF data from the prepreg tape into the simulation software. This makes it possible to accurately describe the anisotropic optical properties of CF/PEEK composites, which greatly simplifies the simulation work.

### 2.3. A Laser Irradiating Thermoplastic Composite Tape

Generally, the methods for measuring the distribution of laser irradiance are categorized as follows: laser ablation, diffuse reflection, calorimetric array, photoelectric matrix method, and so on. The laser-assisted automated tape placement process is a dynamic and complex procedure. Using the previously mentioned techniques, it is difficult to directly measure the interaction between the laser beam and the carbon fiber reinforced thermoplastic composite. The optical model for the lay-up process could not be directly validated, but it might be verified indirectly by comparing the simulated and measured temperature distributions [16,17,22,34,35]. Accordingly, the surface temperature experiment was designed with an inclination of the laser beam to investigate the effect of the curvature feature on the surface temperature, microstructure, and melt pool depth and verify the validity of the theoretical model and the optical-thermal simulation model. Figure 3 shows the schematics of laser irradiance on the surface of the curved tape.

In this study, a laserline GmbH LDM series diode laser with a rectangular-shaped spot (*h*_0_ × *w*_0_) measuring 12 × 14 mm^2^ in the focal distance and a flat top distribution was used as the heat source. A FOTRIC 626 infra-red camera was employed. It has a high sensitivity sensor capable of measuring temperature differences of 1 °C at 30 Hz, at a resolution of 384 × 288 pixels, and the precise temperature measurement range is −20~650 °C, and the temperature averaged over the entire width of the camera image. The DVM-6A 3D microscope was employed to characterize the surface morphology of curved tape after laser heating.

## 3. Basic Principles

### 3.1. Laser Power Density Calculation

An inclination laser beam interaction with the curved CF/PEEK tape can be represented by the absorption behavior and the actual power density distribution. The power density of the laser beam (*E_l_*(*h*)) is described as the ratio of the laser output power (*P_l_*) to the area of the laser spot in the focal distance (*A_l_*), which is given by [14]:(2)$$E_{l}\left(h\right)=\frac{f_{d}P_{l}}{h_{0}w_{0}}$$
where, *P_l_* is the Laser output power, *h*_0_, and *w*_0_ is the height and width of the laser spot, respectively, *f_d_* is the intensity distribution function, and equal to constant when the laser beam was flat top.

### 3.2. Analytical Model

The absorption of the material was not uniform when the laser beam incident angle changed [36]. In the curved tape, the incident angle varies in different positions of the laser beam irradiation, resulting in a non-uniform distribution of the actual power density on the surface. A theoretical model based on energy conservation and specular reflection has been developed. A laser spot is assumed to reflect at most once, in a specular (mirror-like) fashion. The laser beam distribution is assumed to be flat top, which means that *f_d_* is equal to 1. The lay-up procedure is simplified as the geometric model, as shown in Figure 4, and the symbols used in the model are listed in Table 2.

Let *l_r_* be the coordinate along with the tape (*l_r_* = 0 at its origin point) and *h* be the coordinate of the laser spot (*h* = 0 at its midpoint). Based on the simplified geometric model, a relation can now be derived between the irradiated length *l_r_* and the coordinate of the laser spot *h*. The coordinate mapping relation is given by Equation (3).
(3)$$l_{r}=f\left(h\right)=R_{r}\left(\cos^{-1}\left(\cos\alpha_{r}-\frac{h}{R_{r}}\right)-\alpha_{r}\right)$$

When both $h\in \left[0,h_{0}\right]$ and $l_{r}\in \left[0,l_{0}\right]$, these equations are valid only in the irradiated zone shown in Figure 4. Based on the principle of energy conservation, the irradiance of tape (*Er*, irradiance per unit area) can now be related to the radiant (*E_l_*, power density of laser output) of the laser beam. The energy emitted by the height of the laser beam $h\in \left[h_{0},h_{0}+\Delta h\right]$ equals $E_{l}\left(h\right)\Delta h$, ∆*h* is assumed to be infinitesimally small. All of this energy is received by the tape $l_{r}\in \left[l_{0},l_{0}+\Delta l\right]$, resulting in Equation for the tape irradiance.
(4)$$E_{r}\left(l_{0}\right)\Delta l=E_{l}\left(h_{0}\right)\Delta h$$

When taking the limit Δh→0, the resulting relation between irradiance flux (*Er*(*lr*)) and radiant flux (*E_l_(h)*) might be expressed like Equation (5).
(5)$$E_{r}\left(l_{r}\right)=\frac{E_{l}\left(h\right)}{\left(\Delta l_{r}/\Delta h\right)}=\frac{E_{l}\left(h\right)}{f^{\prime }\left(h\right)}=E_{l}\left(h\right)\sin\left(\frac{l_{r}}{R_{r}}+\alpha_{r}\right)$$

According to geometric relations, the angle of incident *θ_ri_* at the tape surface may be written like Equation (6).
(6)$$\theta_{ri}=\frac{\pi }{2}-\frac{l_{r}}{R_{r}}-\alpha_{r}$$

In other words, some of the beams are absorbed and some of it is reflected, and the absorption and reflection behaviors are related to the inc. Let *S*(*θ_ri_*) (Equation (7)) be the fraction of the incident laser beam that is reflected based on the Schlick equations [26],
(7)$$S\left(\theta_{ri}\right)=F_{0}+\left(1-F_{0}\right){\left(1-\cos\left(\frac{\pi }{2}-\frac{l_{r}}{R_{r}}-\alpha_{r}\right)\right)}^{5}$$
where $F_{0}={\left(\left(n_{1}-n_{2}\right)/\left(n_{1}+n_{2}\right)\right)}^{2}$, and *n*_1_, *n*_2_ is the refraction of air and CF/PEEK composite respectively, *n*_2_ = 1.95 [23]. Consequently, the irradiance *E_r_*(*l_r_*) may be split into two terms; a heat flux *q_abs_*(*l_r_*) that is absorbed by the curved tape surface (Equation (8)) and a reflected flux.
(8)$$q_{abs}\left(l_{r}\right)=\frac{f_{d}P_{l}}{h_{0}w_{0}}\left(1-S\left(\frac{\pi }{2}-\frac{l_{r}}{R_{r}}-\alpha_{r}\right)\right)\sin\left(\frac{l_{r}}{R_{r}}+\alpha_{r}\right)$$

### 3.3. Melt Pool Depth Approximation

The validity of the theoretical model is indirectly verified by the melt pool depth of the tape. However, the presence of fibers in CF/PEEK composites makes it arduous to characterize the melt pool depth based on cross-sectional microscopic morphology. According to the analytical approximation for melt pool depth [37], with consideration of the surface heat flux *q_abs_*(*l_r_*) (Equation(8)) and conductivity *k_z_* (Table 1), the approximated analytical solution can be written as Equation (9).
(9)$$\delta_{melt}\approx -k_{z}\cdot \frac{T_{melting}-T_{surface}}{q_{abs}\left(l_{r}\right)}$$
where *T_surface_* is the final surface temperature, *T_melting_* < *T_surface_* < *T_degradation_*.

## 4. Optical-Thermal Model for the Curved Specimen

Trace Pro software was used to establish a three-dimensional ray tracing model for the curved tape to study the irradiance and absorbed heat flux distribution on the surface of the tape and evaluate the theoretical model’s accuracy. Based on micro-surface theory, the scattering model is employed to characterize the anisotropic reflection behavior of CF/PEEK composites. Carbon fiber composite absorption and reflection behaviour occurs at the surface layer. Hence, the anisotropic reflection behaviour of CF/PEEK composites can be represented by the Asymmetric Table BSDF scattering model derived from measurements of the bidirectional reflection distribution function.

Based on the surface heat flux obtained by theoretical and optical models, a thermal model was developed and utilized to investigate the effect of the heat flux distribution on the surface temperature distribution. The curved surface of the heat flux distribution was imported into ANSYS Workbench to analyze the temperature distribution during the laser heating with the material parameters shown in Table 1 and the boundary conditions referenced in [38]. Then, the validity of the optical-thermal coupling model was demonstrated by the experiment of laser heating curved tape (Section 2.3).

## 5. Results and Discussion

### 5.1. Effect of Curvature Feature on the Optical Behavior

Figure 5 shows the variation of reflection patterns with different incident angles. As the beam irradiation zone height increases, its incident angle reduces. A semi-crescent shape appears in the reflection pattern. With the increased distance from the origin, the curvature radius of the semi-crescent becomes smaller, and the scattered light encircles the semi-crescent. In other words, the light is scattered more in the semi-crescent portion. In the origin position, the semi-crescent shape is not noticeable, and the surrounding light is more divergent. The reason is that the beam incident angle at the origin point is larger, the length of the spot illuminated on the surface of the sample along the fiber orientation becomes more divergent, and the extent of fiber bending expands, resulting in more scattered light.

The study [23] revealed that the transmissivity was less than 0.1% at all wavelengths, indicating that the carbon fiber has good absorption. Consequently, the reflection and absorption behavior were merely considered for carbon fiber reinforced thermoplastic composites. Figure 6 depicts the reflectance variation with the incident angle increase obtained by the specular reflection model in the theoretical calculation and reflection measurement. As the incident angle is less than 45°, the reflectance calculated from the specular model is nearly consistent with the experimental data and is approximately 10%. When the incident angle is greater than 45°, the reflectance increases with the increase of the incident angle, the specular model data are slightly higher than the experimental results. When the incident angle is greater than 75°, differences between theoretical analysis and experimental results increase with the increase of incident angle. From Figure 5, it can be seen that the variation of the angle of incidence ranges from 38.5° to 75°. Accordingly, specular reflection could be used to develop the theoretical model.

### 5.2. Effect of Curvature Feature on the Power Density Distribution

The curved tape irradiated by the specific inclination angle of the laser beam and the curvature feature of the CF/PEEK composite tape can result in an inconsistent beam cross-section dimension and non-uniform surface heat flux distribution during the laser heating lay-up process. The variation of irradiated area and irradiance flux density on the surface of the curved tape with the laser beam position, calculated by Equation (5), is depicted in Figure 7. The Irradiated length of the laser beam with the increase of irradiation length at the different irradiated positions for curved tape and flat specimen is shown in Figure 7 a. As the distance from the origin increases, the irradiance on the surface of the tape amplifies, as shown in Figure 7b, while the irradiance on the surface of the flat specimen remains constant, causing a more significant difference in irradiance flux between the flat and the curved tape, and the irradiance in the flat is only one-third of curved tape at the end of the laser beam position.

The described results indicated that the interaction between the laser beam and the curvature feature of tape was more complex, such as the beam elongation, surface irradiance, and incident angle. Consequently, the focus of the following section will be on the interaction of curved tape. The feature of curvature causes the incident angle between the beam and the specimen to increase with irradiation position, which influences the absorption of the tape surface. Based on the relationship between the incident angle and the beam reflection behavior, the influence of distance from the position on incident angle and reflectivity was shown in Figure 8. The curvature features resulted in the incident angle of the beam and the surface absorptance of curved tape ranging from 75° to 38.5° and 0.70 to 0.89, respectively, indicating that the position of the laser beam irradiation away from the origin point is more favorable for absorption during the laser heating lay-up process.

Figure 9a shows the heat flux distribution on the surface of the curved tape. The power density at the origin position was only one-quarter at the end, contributing to the non-uniform distribution of irradiance flux density and absorptance on the curved tape surface. The theoretical model demonstrates the variation of heat flux with the irradiated position on the curved surface, which is obtained by averaging the heat flux across the width direction of the curved tape. Although there is a slight difference between the simulation model and the theoretical model for the surface heat flux close to the origin point, the power density distribution derived from the theoretical model was in good agreement with the simulation model, as shown in Figure 9b. The optical simulation takes into account the effect of anisotropic scattering for carbon fiber reinforced thermoplastic composites, resulting in more diffuse reflection and scattering reflection. The irradiance flux density received by the surface of curved tape was lower, whereas the specular reflection is considered in the theoretical model.

Figure 10 shows the power density distribution obtained by the theoretical model along the length of irradiation on the surface of the curved tape and flat specimen, respectively. There is a significant difference between the curved tape and flat specimen for the power density distribution, and the surface heat flux on the flat specimens is only 1/5 of the radiant laser existence, while it increases to 7/10 of the radiant laser existence in the curved tape for the maximum heat flux. For the flat specimen, the beam is incidental at a large angle, which increases the spot size of the specimen cross-section to 3.86 times the original spot size and reduces the absorptance, resulting in a lower surface heat flux. The curvature feature made the irradiance flux density distribution non-uniform (Figure 7) and the absorptance variant (Figure 8) on the surface. Consequently, the heat flux on the specimen surface increases with distance from the origin, and the heat flux at the origin point is only 25.7% of that at the end position. Hence, the power density distribution in the curved tape was not as uniform as in the flat specimen and existed a meaningful difference during the laser heating lay-up process.

### 5.3. Effect of Power Density Distribution on the Temperature Distribution and Surface Morphology

The laser irradiating thermoplastic composite tape experiment was designed to investigate the effect of non-uniform power density distribution on the temperature distribution, melting pool depth (approximate solution), and surface micromorphology in the laser-irradiated area, as well as verify the validity of the optical-thermal coupling simulation and theoretical models.

The surface temperature distribution on the curved tape derived from the thermal simulation model was compared with the experimental data to validate the accuracy of the simulation model. Figure 11 depicts the surface temperature distribution obtained by experimental measurements and the thermal simulation model. As depicted in the graph, the temperature increases gradually with distance from the origin, which is consistent with the power density distribution obtained by the theoretical model. It was also observed that the surface temperature obtained by the thermal simulation model was lower than the actual measured surface temperature and that the maximum temperature deviated from the position with the highest heat flux. The reason may be the thermal boundary in the end position of the irradiated tape. In addition, the surface temperature obtained from the thermal simulation model was in good agreement with the experimental results, indicating that the thermal simulation model was valid, and the heat flux obtained by the optical model was available. When the irradiation distance is increased to 14 mm, the surface temperature exceeds the melting point (*T_melting_*) of the CF/PEEK composite tape, which means that the composite tape exits a molten state at this position [37]. Consequently, the effect of power density distribution on the laser beam irradiated area was investigated by analyzing the variation in melt pool depth and surface morphology.

Figure 12 depicts the melting pool depth approximation determined by Equation (9) and surface morphology after laser heating for the curved tape. Suppose the laser beam irradiated in length is less than 14 mm. In that case, there is no melting pool depth approximation on the curved tape surface in this area. This indicates that the surface temperature was below *T_melting_* for the curved tape, so the approximate melt pool depth can be assumed to be zero. In addition, there was no apparent variation in the curved tape for the surface morphological feature shown in Figure 12b, which was essentially consistent with the power density, temperature distribution (Figure 11), and the theoretical melt pool depth approximation. The melt pool depth begins to appear when the distance from the origin position exceeds 14 mm (Figure 12a), and the surface morphology made a difference on the curved tape (Figure 12b), as described in the literature [39]. The surface power density and temperature increase with the increase of the laser beam irradiation length along with the curved tape. It causes the approximated melt pool depth to increase from 14.5 μm to 56.5 μm as is shown in Figure 12a, and the variation of melt pool depth is slightly higher than surface morphology characterization. The variation of melt pool depth approximation with distance from the origin obtained from the theoretical study and surface morphology are generally consistent.

### 5.4. Influence of the Intensity Distribution of Laser Beam on the Heating Effect

Given all that, the curvature feature of curved tape resulted in a non-uniform power density distribution on the surface. The farther away from the origin point, the greater the difference in the power density between the curved tape and the flat specimen. According to the character of power density distribution on the curved tape, the intensity distribution of the Super-Gaussian beam profile in the height direction is proposed to improve the uniformity of power density distribution on the curved tape and flat specimen for the laser heating lay-up process.

The bidirectional flat-top distribution beam profile with uniform energy distribution could ensure uniform power density distribution in the width direction of the curved tape. However, the curvature feature in the height direction results in non-uniform heat flux distribution along with the length of the curved tape. Consequently, the height direction of the Super-Gaussian beam profile was employed to compensate for the non-uniform heat flux distribution caused by a significant difference in absorptance and irradiance flux density.

When a pumped beam has a non-Gaussian profile, the distribution function of the laser beam profile can be modified by different exponents [40]. This model, known as the Super-Gaussian distribution, is described by the following equation:(10)$$q\left(x,y\right)=q_{0}\exp\left(-a\left({\left(\frac{x}{h_{0}}\right)}^{m}+{\left(\frac{y}{b_{0}}\right)}^{n}\right)\right)$$
where *x* and *y* are actual coordinates of a material point, *m* and *n* are exponents (for *m* = *n* = 2 the distribution is the typical Gaussian distribution).

According to the description in Equation (10), as *n* approaches infinity, the laser spot is uniformly distributed, allowing for uniform heating of the curved tape in the width direction. If the order *m* approaches infinity, the laser beam becomes the flat top distribution. With the decrease of the exponent *m*, the laser beam changes from a flat top distribution to a typical gaussian distribution, as shown in Figure 13a. When the order *m* is equal to 2, the highest intensity distribution occurs in the center of the laser beam. As *m* approaches infinity, the intensity distribution is uniform. Figure 13b shows the surface heat flux distribution variation at the Super-Gaussian beam profile with different (even) exponents on the curved tape. A Super-Gaussian beam was employed. Through increasing the distance from the origin, surface power density increased and then decreased compared to the flat-top distribution. The smaller the exponent m is, the lower the heat flux of the curved surface is. When the beam profile distribution changed from flat top to typical Gaussian, the position of the highest heat flux appearance moved from 22 mm to 12 mm on the curved surface, and it approached the origin point of the curved tape. Above all, the surface heat flux distribution is more homogeneous when the typical Gaussian beam profile rather than the Super-Gaussian beam profile is used for the curved tape, and the irradiance inhomogeneity increases with the order m increases in the Super-Gaussian function.

The effect of the intensity distribution of the Super-Gaussian profile beam on the surface temperature distribution is depicted in Figure 14. When the flat top distribution (m→∞) beam was utilized, the high-temperature area on the surface was the farthest distance from the origin point. As the exponent m decreased, the distance between the high-temperature region and the origin point reduced, and the region of the curved tape with the highest temperature moved toward the origin point. A typical Gaussian profile beam was employed, and the high-temperature zone moved from 18~22 mm to 9~15 mm, and was closer to the origin position. In addition, the width of the high-temperature zone was greater than that of the Super-Gaussian (*m* ≥ 4) beam. The main reason for improving the uniformity of the irradiance on the curved tape was that the curvature feature decreases the power density with the increase of the irradiated length (Figure 10), and the relative intensity reduced by increasing the height of the laser beam for a typical Gaussian profile beam. This could compensate for the variation in irradiance and absorptance, resulting in a more uniform heat flux distribution. Consequently, a typical Gaussian beam was more appropriate for the laser heating lay-up process.

The typical Gaussian profile beam can reduce the impact of the curvature feature on power density distribution and improve irradiation uniformity. However, the effect of intensity distribution on the power density distribution of the flat specimen required further investigation. Figure 15 shows the power density distribution using a typical Gaussian profile beam on curved tape and flat specimens. It is evident that the difference in power density between the curved tape and the flat specimen was significant when the flat top profile beam was employed, and the highest intensity was only one-fourth that of the curved tape. When the order *m* decreased, the difference in power density decreased, and a typical Gaussian profile beam had a minimum difference of approximately half of that of curved tape. Therefore, a typical Gaussian profile beam significantly improved the irradiation uniformity during the laser heating lay-up process.

The influence of the curvature radius of the tape and the inclination of the laser beam on the power density distribution is shown in Figure 16. As the curvature radius increases, the irradiated length increases, and the power density at the origin increases from only 1/5 to 1/3 of the highest power density. Moreover, the uniformity of irradiance on the surface was improved. Increasing the beam tilt angle modified the power density distribution, as depicted in Figure 16b. The power density at the origin position and the power density distribution on the surface gradually improved, resulting in a more uniform power density distribution, as depicted in Figure 16b. The uniformity of heat flux distribution could be improved by regulating the laser beam’s curvature radius and inclination angle. However, these parameters influenced the shadow area [8]. Hence, the curvature radius and inclination must be properly considered.

## 6. Conclusions

In this paper, reflection behavior, power density distribution, and surface morphology were employed to study the impact of the curvature on the laser beam interaction across the CF/PEEK composites and the influence of relative intensity distribution, curvature radius, and inclination angle on the uniformity of irradiance was investigated as well.

It was found that the curvature feature of the CF/PEEK composite indeed affected the optical behavior. The curvature feature made the incident angle range from 75° to 38.5°, and as the irradiated position was farther away from the origin, the crescent reflection pattern became gradually more apparent.Theoretical calculation results show that the curvature feature of the income tape makes the irradiance and absorptance change, causing the power density to increase as the distance from the origin increase. Hence the minimum energy density occurs at the origin position, the power density was only 18% of the radiant laser existence. The theoretical model was in good agreement with the optical simulation model. Accordingly, the difference in power density distribution between the income tape and the flat was great.The uneven distribution of power density caused a more significant change in the temperature distribution, melt pool depth, and surface morphology along the circumference. Experimental results indicated that the surface temperature variation was consistent with the power density distribution. Moreover, the melt pool depth appeared at 14 mm from the origin, and as the distance increased, the approximated melt pool depth increased to 56.5 μm. It also implied that a remarkable change in surface morphology occurred in the curved tape.The Super-Gaussian function was introduced into the theoretical model to describe the relationship between exponent and power density distribution. As the exponent *m* decreased, the irradiation uniformity improved, and the location of the highest power density became closer to the origin. When the typical Gaussian profile beam in the height direction of the beam is employed, the higher relative intensity compensates for the energy loss due to the lower irradiance and absorptance at the origin. Consequently, the resulting irradiation uniformity increased by up to 25%, significantly reducing the difference between the curved tape and the flat during the laser heating lay-up process.

In short, this study provides a fundamental understanding of the possible influences of curvature features on laser heating from a laser-materials interaction perspective. This will allow the researchers to design strategies such as the alteration of the intensity distribution of the laser profile, curvature radius, and inclination angle of the laser beam to compensate for the variation of the irradiance and improve the quality of laser heating lay-up additive manufacturing parts in aerospace applications.

## Figures and Tables

**Figure 1 polymers-15-00289-f001:**
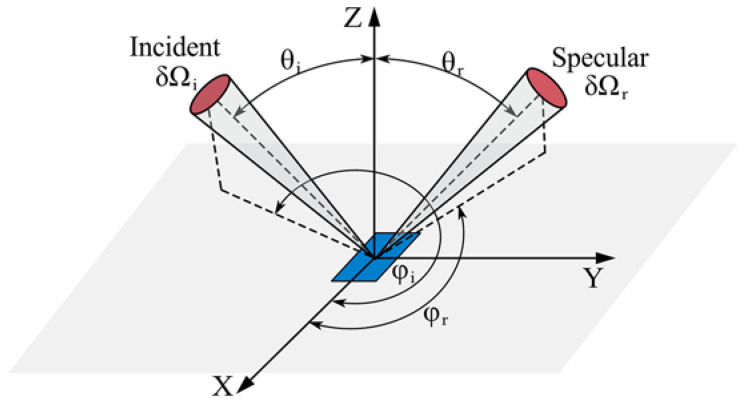
The definition of Bidirectional Reflection Distribution Function.

**Figure 2 polymers-15-00289-f002:**
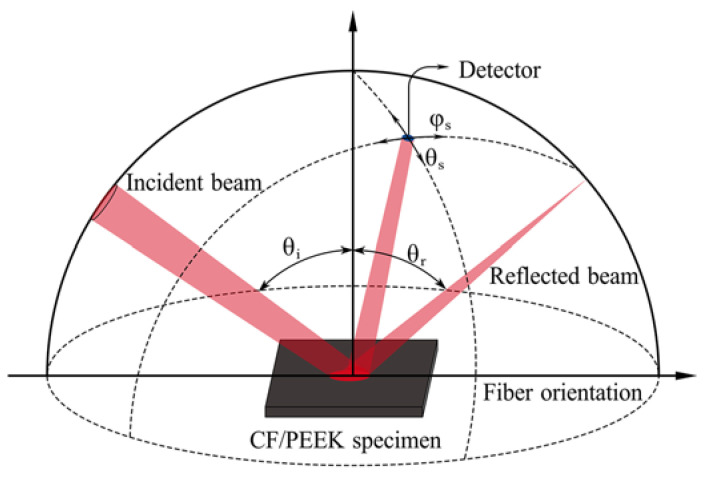
Illustration of the BRDF measuring system for the CF/PEEK composite specimen.

**Figure 3 polymers-15-00289-f003:**
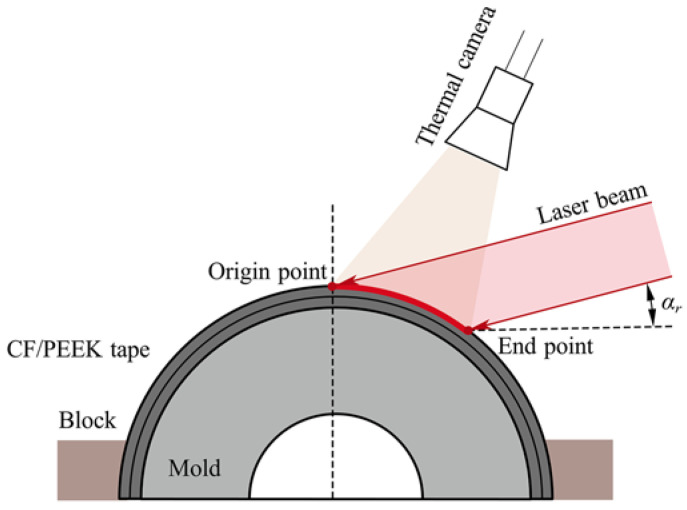
Experimental setup to measure temperature during laser heating the CF/PEEK tape.

**Figure 4 polymers-15-00289-f004:**
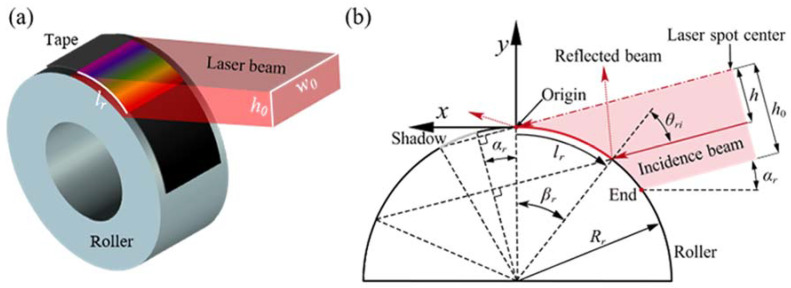
Laser irradiation model (**a**) and simplified 2D geometry for analytical model (**b**).

**Figure 5 polymers-15-00289-f005:**
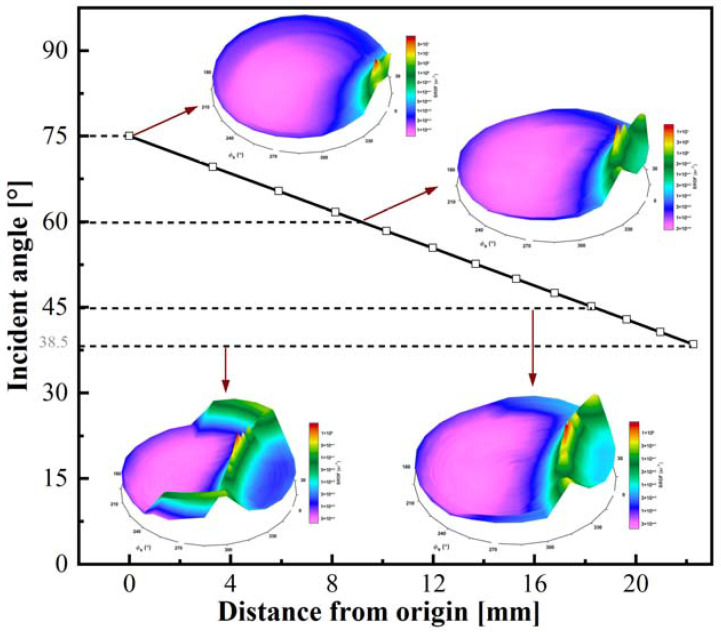
The variation of reflection pattern and inclination angle α_r_ in a different position.

**Figure 6 polymers-15-00289-f006:**
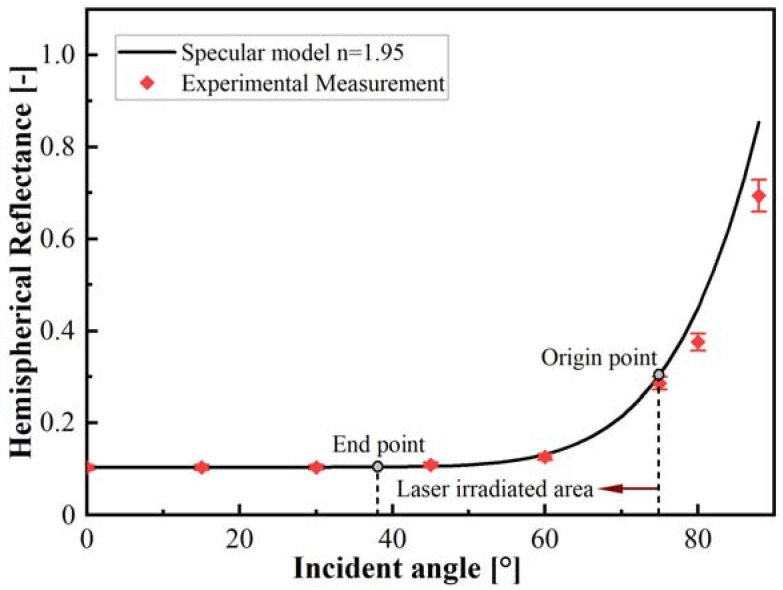
Angular dependence of reflectance for CF/PPEK composite.

**Figure 7 polymers-15-00289-f007:**
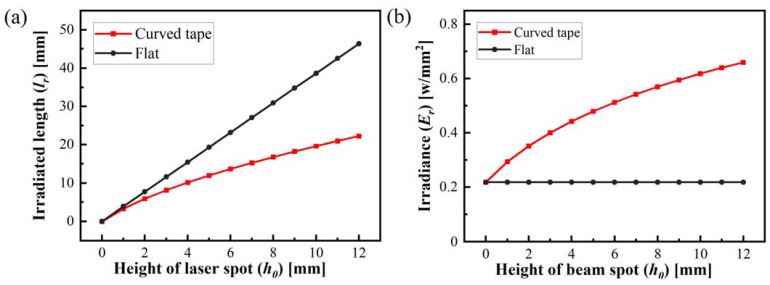
Quantitative graphs of irradiated length (**a**) and irradiance (**b**) variations based on Equations (4) and (5).

**Figure 8 polymers-15-00289-f008:**
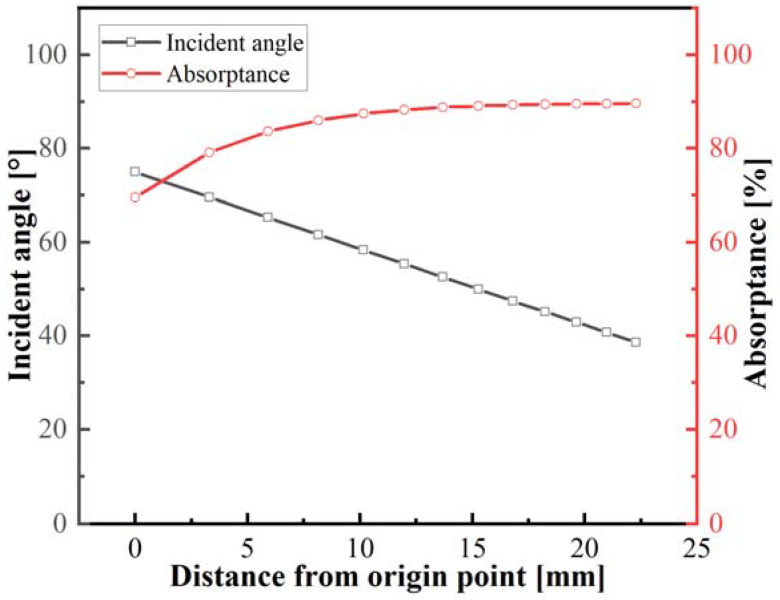
The effect of distance from the origin on incident angle and absorptance.

**Figure 9 polymers-15-00289-f009:**
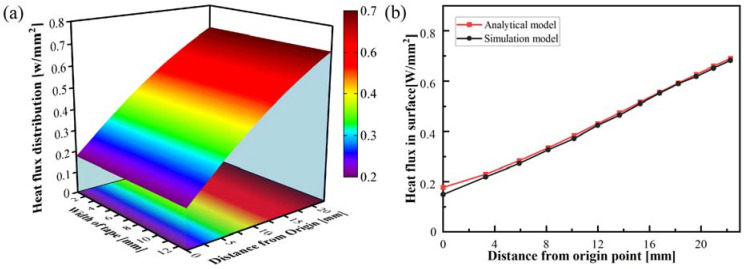
Heat flux distribution obtained by the theoretical model (**a**) and comparison with simulation model results (**b**).

**Figure 10 polymers-15-00289-f010:**
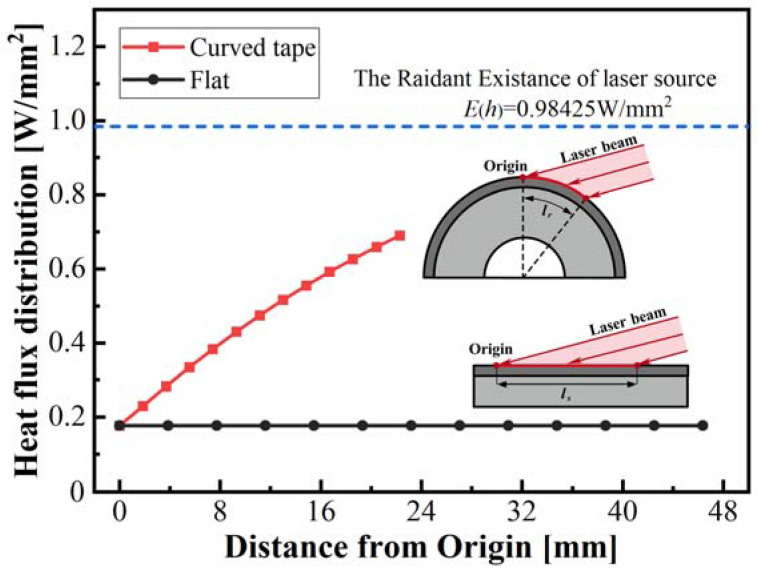
Comparison of heat flux distribution in curved tape and flat specimen.

**Figure 11 polymers-15-00289-f011:**
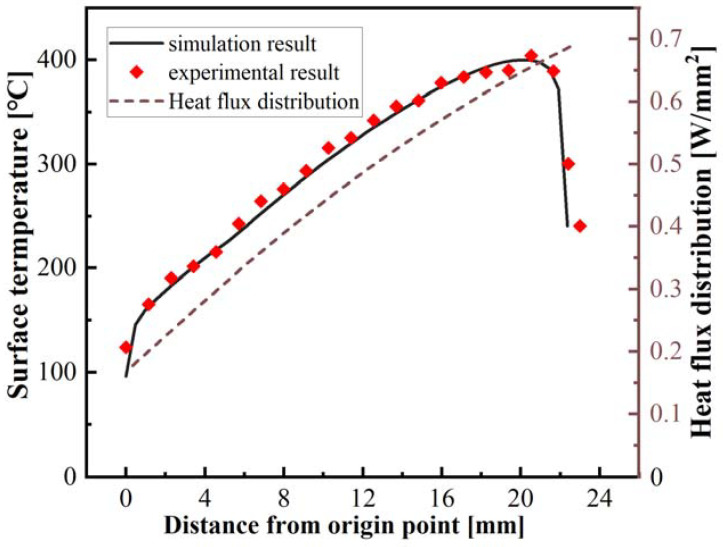
The temperature distribution on the tape surface (comparison of experiment and simulation results).

**Figure 12 polymers-15-00289-f012:**
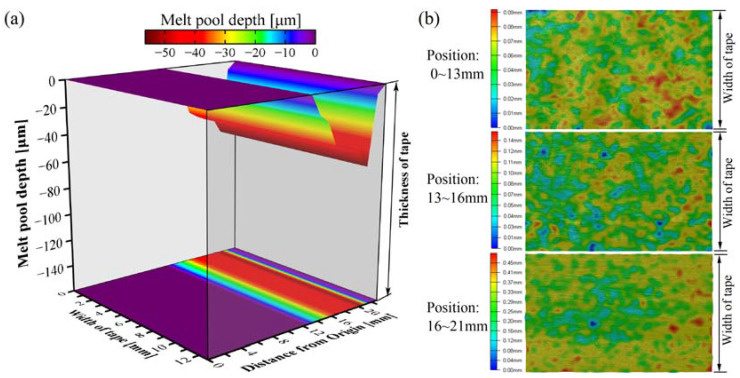
Melt pool depth approximation (**a**) and surface morphology (**b**) after laser heating.

**Figure 13 polymers-15-00289-f013:**
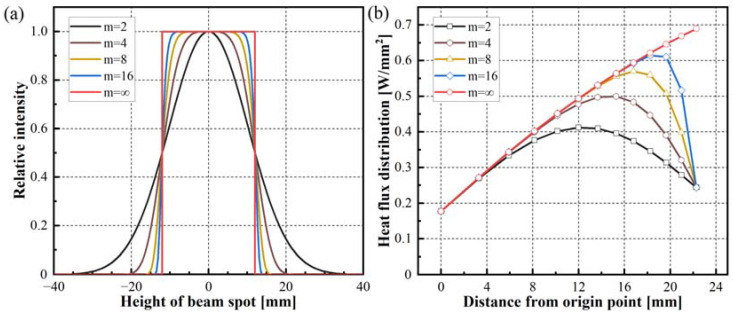
Intensity normalized distribution of the Super-Gaussian profile beam (**a**) and effect on the surface heat flux distribution on the curved tape (**b**).

**Figure 14 polymers-15-00289-f014:**
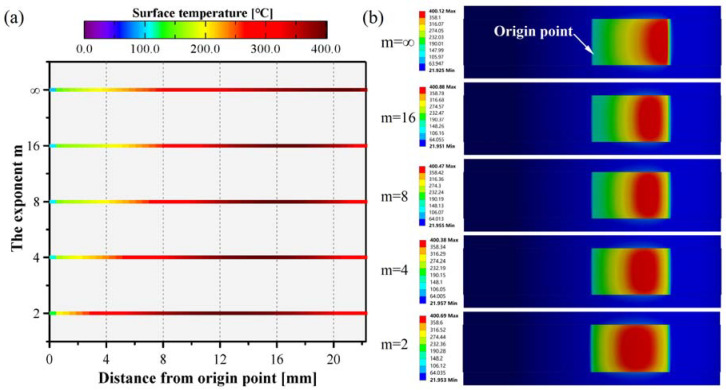
The center temperature in width direction (**a**) and surface temperature distribution (**b**) of the curved tape was obtained by the thermal simulation model.

**Figure 15 polymers-15-00289-f015:**
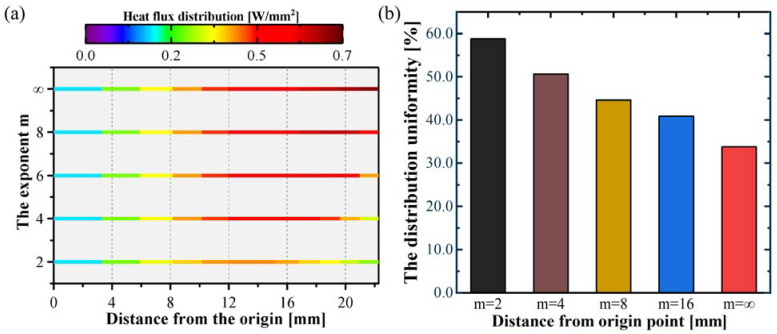
The influence of the exponent m on the heat flux distribution (**a**) and the distribution uniformity (**b**).

**Figure 16 polymers-15-00289-f016:**
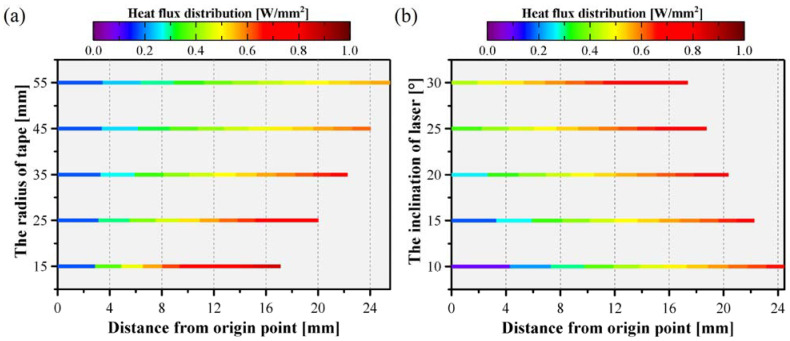
The effect of curvature radius (**a**) and inclination of the laser beam (**b**) on the power density distribution.

**Table 1 polymers-15-00289-t001:** Material constants of the CF/PEEK composite.

Parameters	Melting Temperature *T*_melting_ [°C]	Density ρ [kg/m^3^]	Specific Heat Capacity C_p_ [J/kg/K]	Thermal Conductivity	
				k_xx_ [W/m/K]	k_yy_(k_zz_) [W/m/K]
Value	343	1540	1100	4.92	0.61

**Table 2 polymers-15-00289-t002:** Geometric parameters.

Parameters	Symbol	Value
Laser power	*P_l_*	150 W
Laser spot width	*w* _0_	14 mm
Laser spot height	*h* _0_	12 mm
Laser beam inclination angle	*α_r_*	0.26 rad
Tape radius	*R_r_*	35 mm
Tape width	*w_t_*	12.7 mm
Roller width	*w_r_*	20 mm

## Data Availability

Not applicable.

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
