# Peer review of "Influence of Curvature Feature on Laser Heating during Tape Placement Process for Carbon Fiber Reinforced Polyether Ether Ketone Composite"

_polymers, 2023, doi:10.3390/polym15020289_

Round 1

Reviewer 1 Report

The research article ‘Influence of curvature feature on laser heating during tape placement process for carbon fiber reinforced polyether etherketone composite’ analyzes the influence of curvature characteristics on the reflection behaviour, the heat flux distribution, and the microstructure of the irradiated surface during the laser heating lay-up process laser heating process and the interaction cross-section. The article aims to contribute to offering a fundamental base for the subsequent improvement of the temperature distribution.

The manuscript is well written and correctly analysed. Only  a couple of observations are necessary prior to publication

1.      Eliminate ‘of’ from the sentence in line 30

2.      Review the spelling of the sentence. Lines 90-95.

3.      ‘The CF/PEEK composite prepreg tape applied in the study was manufactured by Evonik Industries AG…’ The sentence is unclear. So the composite is not commercial? Did the authors pay the Evonik company to make this unique composite for this research article? Should Evonik Industries AG be in the acknowledgments of this research article? The authors should review and modify the sentence.

Author Response

Dear Reviewers,

Comments:

Thank you very much for your time involved in reviewing the manuscript and your very encouraging comments on the merits.

We also appreciate your clear and detailed feedback and hope that the explanation has fully addressed all of your concerns. In the remainder of this letter, we discuss each of your comments individually along with our corresponding responses.

To facilitate this discussion, we first retype your comments in italic font and then present our responses to the comments.

Comment 1:

Eliminate ‘of’ from the sentence in line 30.

Response 1:

Thank you for you careful review. We have eliminated ‘of’ from the sentence in line 30.

Comment 2:

Review the spelling of the sentence. Lines 90-95.

Response 2:

Thanks for your careful checks. We are sorry for our carelessness. Based on your comments, we have corrected the spelling of the sentence, Lines 90-95, and also made the corrections in the manuscript using revision mode.

Comment 3:

The CF/PEEK composite prepreg tape applied in the study was manufactured by Evonik Industries AG…’ The sentence is unclear. So the composite is not commercial? Did the authors pay the Evonik company to make this unique composite for this research article? Should Evonik Industries AG be in the acknowledgments of this research article? The authors should review and modify the sentence.

Response 3:

We sincerely thank the reviewer for careful reading. This material has been already commercially available, and supplied by Evonik Industries AG (VESTAPE®).As suggested by the reviewer, we have made the corrections to make the sentence better understood and express the meaning accurately. Moreover, we has carefully modified the sentence in the manuscript through revision mode. We were really sorry for our unclear expression. Thank you for your reminder again.

We would like to thank the referee again for taking the time to review our manuscript.

Sincerely,

The Authors

Reviewer 2 Report

The manuscript “Influence of curvature feature on laser heating during tape placement process for carbon fiber reinforced polyether ether ketone composite” is included in the topic of the Journal Polymers.

The introduction of this article provide background and presents the effect of curvature features on the reflection behaviour, heat flow distribution, and microstructure of the surface of the laser beam irradiation area is investigated during the laser heating lay-up process, and the interaction cross-section of the laser beam on the income tape is obtained to contribute to offering a fundamental basis for the subsequent improvement of the temperature distribution.

Please relate some studies for the polymers composed from carbon fiber reinforced polyether ether ketone composite.

Materials and methods are good described in order to identify the methods for the study.

 In Results and Discussions, please make a comparation between theoretical study and practical/experimental studies.

Author Response

Dear Reviewers,

Comments:

Thank you very much for your time involved in reviewing the manuscript and your very encouraging comments on the merits.

We also appreciate your clear and detailed feedback and hope that the explanation has fully addressed all of your concerns. In the remainder of this letter, we discuss each of your comments individually along with our corresponding responses.

To facilitate this discussion, we first retype your comments in italic font and then present our responses to the comments.

Comment 1:

Please relate some studies for the polymers composed from carbon fiber reinforced polyether ether ketone composite.

Response 1:

We are grateful for the suggestion. Before we conducted this study, a full literature review was performed and these studies cited in the introduction primarily relates to carbon fiber reinforced polyether ether ketone composite, however, since high-performance thermoplastic composites like carbon fiber reinforced polyether ether ketone composite are also referred to as thermoplastic composites, this article did not take this section of the studies on the polymers composed from carbon fiber reinforced polyether ether ketone composite into account. The reviewer’s suggestion made me realize the importance of these studies, we have checked the literature carefully and refined these studies on the polymers composed from carbon fiber reinforced polyether ether ketone composite in the INTRODUCTION part in the revised manuscript.

Comment 2:

Materials and methods are good described in order to identify the methods for the study. In Results and Discussions, please make a comparation between theoretical study and practical/experimental studies.

Response 2:

We appreciate the reviewer’s positive evaluation of our work. According the suggestion, we have reviewed carefully and made a comparation between theoretical study and practical/experimental studies. In the revised version of the manuscript, we have refined the Result and Discussion sections (especially the comparation between theoretical study and practical/experimental studies), for example, we have (1) restructured the results (figures) to make the comparation between theoretical study and practical/experimental studies clearer; (2) added the result comparisons into the Result and Discussion (5.1 section, 5.2 section, and 5.3 section), these were as follows.

a) In section 5.1. A comparison of the theoretical analysis and experimental results has been refined in line 250-254.

b) In section 5.2. Since it was challenging to measure the actual heat flux distribution, it is preferable to determine the variation of surface heat flux through surface temperature distribution. As a result, the practical studies was optical simulation, and the comparation has been present in line 292-299.

c) In section 5.3. the comparison between theoretical and experimental studies can only be made indirectly; this is done in the paper in lines 326-327. Then, we added the comparation between melt pool depth approximation (theoretical) and surface morphology characterization (experiment) in lines 345-361.

We would like to thank the referee again for taking the time to review our manuscript.

Sincerely,

The Authors